# Optimal LiDAR Data Resolution Analysis for Object Classification

**DOI:** 10.3390/s22145152

**Published:** 2022-07-09

**Authors:** Marjorie Darrah, Matthew Richardson, Bradley DeRoos, Mitchell Wathen

**Affiliations:** 1Mathematics Department, West Virginia University, Morgantown, WV 26506, USA; 24D Tech Solutions, Inc., Fairmont, WV 26554, USA; mrichardson@4dtechsolutions.com (M.R.); bderoos@4dtechsolutions.com (B.D.); 3Army Research Laboratory, Adelphi, MD 21005, USA; mitchell.s.wathen2.civ@army.mil

**Keywords:** LiDAR, convolutional neural network, optimal data resolution, simulated data

## Abstract

When classifying objects in 3D LiDAR data, it is important to use efficient collection methods and processing algorithms. This paper considers the resolution needed to classify 3D objects accurately and discusses how this resolution is accomplished for the RedTail RTL-450 LiDAR System. We employ VoxNet, a convolutional neural network, to classify the 3D data and test the accuracy using different data resolution levels. The results show that for our data set, if the neural network is trained using higher resolution data, then the accuracy of the classification is above 97%, even for the very sparse testing set (10% of original test data set point density). When the training is done on lower resolution data sets, the classification accuracy remains good but drops off at around 3% of the original test data set point density. These results have implications for determining flight altitude and speed for an unmanned aerial vehicle (UAV) to achieve high accuracy classification. The findings point to the value of high-resolution point clouds for both the training of the convolutional neural network and in data collected from a LiDAR sensor.

## 1. Introduction

Processing data collected from an unmanned vehicle’s LiDAR sensors so that decisions can be made is a challenging problem. The trade-offs become: (1) the total area that needs to be mapped, and (2) the size and physical characteristics for which detection or classification of objects is required, and (3) the time it will take to process the data, ideally to allow detection and classification to occur during flight. This research investigates the collection and processing of 3D LiDAR data for object classification and the resolution needed to detect and accurately classify objects in the data. The trade-offs between the resolution of the data and the accuracy of the classification are discussed. The data set used to test these trade-offs is a set of 3D LiDAR data collected by an unmanned aerial vehicle (UAV) (hexrotor) carrying the RedTail RTL-450 LiDAR sensor. The analysis method discussed is a convolutional neural network trained on various objects to be classified in the LiDAR data set.

Convolutional neural networks (CNNs) are one of the most frequently used deep learning (DL) methods and are effective for classifying 3D objects in point clouds. CNNs have been applied to 3-D data sets [1] and specifically to LiDAR data [2]. Prokhorov [3] was one of the first to investigate the application of CNNs for 3D object recognition and classification. In his work, Prokhorov transformed the initial point cloud data into a 3D grid using a binning operation. His CNN consisted of one convolutional layer, a pooling layer, two fully connected layers, and a two-class output layer.

Qi, Su, Mo, and Guibas [4] introduced PointNet, a unified architecture that directly takes point clouds as input, respects the permutation invariance of the points, and outputs class labels. This network applies input and feature transformations and then aggregates point features by max pooling. Kowalczuk and Szymanski [5] employed PointNet deep learning neural network to classify 3D LiDAR data. They found that height above the ground has a big impact on the accuracy of the classification and suggest that dividing the classification process into two stages, basic and other, would be helpful. Wang et al. [6] propose a neural network module dubbed EdgeConv suitable for CNN-based classification that acts on graphs dynamically computed in each layer of the network. This CNN can be plugged into existing architectures. He et al. [7] combined a CNN with morphological profiles (MPs) and a spatial transformation network (STN) [8] to develop a classification framework for LiDAR data that produced excellent results on two test sets. The STN transforms the input data by rotating, scaling, and translating, which assists with extracting spatial information by the CNN.

Another CNN that has distinguished itself by being able to extract features from volumetric data is VoxNet [9], created by Maturana and Scherer. The input to the algorithm is the intersection of a point cloud and a bounding box that may contain clutter. The output is the object class label for this segment of the full data set. VoxNet has an input layer, two convolutional layers, a max-pooling layer, fully connected layers, and an output layer. Maturana and Scherer [9] created a 3D grid representation with three types of grids, a binary grid, a density grid, and a hit grid. Hackel et al. [10] introduced a 3D benchmark dataset and presented various models of CNNs. In one of his models, he generated five independent global 3D grids using different resolutions for each instance. This network had five CNN layers and two fully connected layers. The VoxNet CNN was employed in this research and will be explained in more depth later in the paper.

Many articles have been written that discuss the resolution of point cloud densities needed to accomplish various tasks. When considering the effects of resolution on the accuracy of a digital elevation map (DEM), Lui et al. [11] considered reduced datasets to determine the accuracy of producing corresponding DEMs with 5m resolution. They produced a series of datasets with different data densities, representing 100%, 75%, 50%, 25%, 10%, 5%, and 1% of the original training dataset. Results showed no significant difference in DEM accuracy if data points were reduced to 50% of the original point density. However, the processing time for DEM generation with the 50% data set was reduced to half the time needed when using the original 100% dataset.

Peng et al. [12] investigated point cloud density and its effects on the accuracy of determining tree height in tropical rain forests. He and his team collected LiDAR data at a consistent flight altitude of 150 m, and then down sampled to obtain five different point cloud densities (12, 17, 28, 64, and 108 points/m^2^). They developed a canopy height model (CHM) of the trees based on the down-sampled data. They found that with the increase in the point cloud density, the accuracy of the tree height increased for both broadleaf and coniferous trees (108 points/m^2^ produced the best results). For the broadleaf trees, the accuracy (measured in root mean square error) increased by 6.92% as the point cloud density changed from 12 points/m^2^ to 17 points/m^2^ but increased by less than 1% as the point cloud density changed from 17 points/m^2^ to 108 points/m^2^. The results were somewhat different for the coniferous trees, with the accuracy continually increasing from 12 points/m^2^ to 108 points/m^2^. Their research concluded that the lowest LiDAR point cloud density required for sufficient accuracy for tree height extraction was 17 points/m^2^. The teams suggested that this should help researchers formulate forest resource survey plans. 

Błaszczak-Bąk et al. [13] presented a data reduction method, Optimum Dataset (OptD), for pre-processing large LiDAR datasets to reduce processing time and provide optimal object detection. Unlike many data reduction methods, they show the OptD pre-processing step tremendously reduces the dataset size while keeping most of the geometric information of interest for the considered application. Their results show that the OptD method performs much better than random reduction when the original data is reduced by both methods to 1% of the original size. 

Tomljenovic and Rousell [14] extracted objects from Airborne Lasar System (ALS) point cloud datasets. They employed the framework of Cognitional Network Language, a part of the eCognition software package. They converted the ALS data set to the point cloud density. They found that high point density (18 points/m^2^) along with very high resolution (<0.25 m) provided increased accuracy for the extraction algorithm. Lower point cloud densities (7–16 points/m^2^) with lower resolution (0.50 m) provided a stable accuracy. They concluded that since the resulting outcome of the two resolutions showed no significant change in accuracy, either could be used for their purposes. 

This paper explores the idea of testing a CNN for the accuracy of classification of objects in LiDAR data when the data set is reduced to a percentage of the original resolution. First, the methods are tested on a widely available data set, Sydney Urban Objects Dataset (available at https://www.acfr.usyd.edu.au/papers/SydneyUrbanObjectsDataset.shtml (accessed on 1 May 2022)). Then these results are compared with a high-resolution data set that aligns with collecting data at specific altitudes and speeds using the RedTail RTL-450 LiDAR system mounted on a quadcopter (Figure 1). It should be noted that we do not consider environmental factors of the flights in this paper, since when using a small UAV, as pictured in Figure 1 for data collection, conditions must be favorable (low wind and no rain) before flights are attempted. We explain the method used to determine the resolution of each sample, the method for supplementing the high-resolution data set, and then the techniques used for object detection and classification. The results of object classification with various data resolution reductions are presented for both the Sydney Urban Objects Data Set and the high-resolution data set.

## 2. Methods

The methods employed in this research include first completing a trade-off study to determine the resolution of data collected with the RedTail RTL-450 LiDAR system with various operational parameters. This initial part of the study is important to practitioners to assist in the determination of how to collect data with the necessary resolution for optimal classification results. Other researchers conduct data resolution reduction studies [11,12,13,14], and Peng et al. [12] mention the altitude for collecting the original set, but most studies neglect to mention any other parameters of how the data was collected. Other studies also fail to mention how lower resolution can be matched to different flight parameters. Next, we examine the data sets used for training and testing classification with the VoxNet CNN [9]. The CNN is first tested on the Sydney Urban Objects Data Set (collected using the Velodyne HDL-64E LIDAR system). The resolution of this original data is examined and then reduced to determine classification results. Next, the higher resolution data collected using the RedTail RTL-450 LiDAR system is examined, and a method for supplementing the data set is presented since the original data did not contain a sufficient number of samples for training and testing. The resolution of this second data set is reduced, and classification results are presented.

### 2.1. Lidar Data Collection and Trade-Offs

When determining the theoretical point density of a data set, there are several trade-offs to be made. For a LiDAR sensor, the following variables are considered when assessing the quality of a point cloud for classification purposes: (1) points per scan line, (2) scan lines per second, (3) points per second (points/s), (4) scan angle (deg), and (5) beam divergence (mrad).

With the values above, the points per square meter (points/m^2^) are determined by the vehicle’s altitude and flight speed. 

The goal is to optimize the data collection for object detection and classification. For example, if we set the the RTL-450′s operating variables equal to the following, the resultant point densities can be derived as shown in Figure 2 and Table 1. It should be noted that the settings points per second (pulse repetition rate) = 200,000 and scan angle (deg) = 40 were used.

Figure 2 and Table 1 summarize the results when the values above are used. Similar studies were completed with other values.

### 2.2. Input Data Sets and Resolution

#### 2.2.1. Sydney Urban Data Set

The Sydney Urban Objects Dataset is a publicly available LiDAR dataset containing urban objects. Data was collected using a Velodyne HDL-64E LIDAR system. For our study, we selected 14 objects of interest. Before training and testing VoxNet on the dataset, the surface point density was calculated for each object in the dataset. 

CloudCompare v2.11, EDF Group, Paris, France [15] was used to compute the surface density for each point cloud used for training and testing. The surface density is the number of neighbors divided by the neighborhood surface = *N*/(*π* * *r*^2^), resulting in points/m^2^. For our calculations, we used a radius *r* of one meter. After extracting the surface density at each point, the mean density and standard deviation of the entire point cloud were calculated. The mean plus one standard deviation was used as the representative density for each point cloud. Point clouds with similar densities were found or created for each object. The overall average density of the dataset was 61.2 points/m^2^.

#### 2.2.2. RedTail LiDAR System Data Set

To test the accuracy of the implemented CNN on data collected using the RedTail RTL-450 system, a data set containing five specific construction site objects was assembled. Figure 3 [16] below shows an image of high-resolution data collected by the RedTail Sensor over an actual construction site. The data used for training and testing was much lower resolution than what is depicted in this image. This point cloud is included to show the capability of the RTL-450 LiDAR system.

A portion of the data set was collected using a quadrotor carrying the RedTail LiDAR system, and a portion of the data set was simulated to match the collected data. It should be observed that the point cloud models of these objects were from a perspective equivalent to the UAV flying directly above the objects, and thus the top portion is all that is represented. 

Figure 4 shows point clouds collected by the LiDAR sensor used in the training or testing. The colors in the image denote surface density, which is impacted by contours, the reflectivity of the object, and the range of the sensor. 

To increase the size of the training set, simulated data was constructed using 3D models of objects obtained online from various websites (e.g., CGTrader). The objects were downloaded in an object mesh format (stl, obj, fbx). Points on the surface of the mesh were randomly sampled with a specific density that matched the data collected by the LiDAR sensor. The surface density of the point cloud was then checked and compared against the point clouds collected by the LiDAR system. Figure 5 shows a histogram of the density for a particular excavator. From this we can tell the average and standard deviation of the densities. We used the average plus one standard deviation to represent the density of the object. We separated the total data into a training set containing 650 point clouds of the five objects, and the testing set contained 234 point clouds of the five objects.

For example, there were 91 dump trucks with surface densities ranging from 562 to 961 points/m^2^. The average of these is shown in the table below. These densities can be compared with the densities in Table 1 to approximate the altitude and flight speed of the UAV for such data to be collected. For the average densities in Table 2, we can approximate an altitude of 60 m and a flight speed of 6 m per second.

### 2.3. Data Analysis

#### 2.3.1. Detection of Objects within a Larger Set

To detect objects in a larger data set (LiDAR scene), the Hierarchical Grid Model method [17] is employed. This method provides robust 3D object detection from strongly inhomogeneous density point clouds in challenging and dense environments. The approach uses a coarse and dense grid resolution. It starts with a coarse or simple grid model that fits a 2D grid onto the plane *P_z_* = 0, the sensor’s vertical axis in the *z*-direction, and the sensor height as a reference coordinate. It then assigns each point in the 3D point cloud to the corresponding cell, which contains the projection of *p* onto *P_z_* = 0. It stores the cell density, height coordinates, and height properties (max, min, average) within each cell to be used later in the point cloud segmentation. The coarse grid is used for the rough estimation of 3D “blobs” in the scene. This way, the size and location of possible object candidates can be roughly estimated.

Next, we visit every cell in the coarse grid and consider its 3 × 3 neighborhood. For each cell in the neighborhood, consider the maximal elevation and the point cloud density for the cell. To find connected 3D blobs, merge cells where the difference between the maximal point elevation within the target cell and its neighboring cell is less than a pre-defined value. If the criterion is met, we assume the target cell and its neighbor belong to the same object.

After this, we perform a detection refinement step by creating a dense grid subdividing the coarse grid cells where objects are found into smaller cells. The elevation-based cell-merging criterion on the coarse grid level may merge nearby and self-occluded objects into the same blob. This issue can be handled by measuring the point density in each of the dense grid subcells. Nearby objects can be separated, and an empty border cell can be illuminated at this step. 

#### 2.3.2. Classification of Objects

After objects were detected as described above, VoxNet CNN [9] was used for classification. According to Maturana and Scherer [9], there are two components to the VoxNet structure, a volumetric grid representing spatial occupancy and a 3D CNN that predicts a class label from this occupancy grid. To start the process, the point cloud data retrieved from the LiDAR dataset is converted into voxels. Then occupancy grids are developed to represent the state of the environment and maintain a probabilistic estimate of occupancy based on prior knowledge. There are three different occupancy grids developed, the binary grid, density grid, and the hit grid. In the binary grid, every voxel is assumed either occupied (1) or unoccupied (0). In the density grid, each voxel has a continuous density based on the probability of the voxel blocking a sensor beam. The hit grid only considers hits and does not consider unknown and free space. The process of transforming a point cloud into a voxel grid is essentially a coordinate transform of each point in the point cloud, which means this process has complexity O(n), where n is the number of points in the point cloud. This research was performed using a binary grid model. 

The CNN consists of several layers, input layer, convolutional layer, pooling layer, and fully connected layer. The input layer accepts occupancy grids. In our study, we used a 24 × 24 × 24 m gird (32 × 32 × 32 after zero-padding) and individual voxel resolution of 0.4 × 0.4 × 0.4 m. In the pooling layers, the input volume is downsampled by a factor of *m* along each spatial dimension. And finally, in the fully connected layer, there are n output neurons. Each neuron is a learned linear combination of all outputs from the previous layer passed through a nonlinearity. For the final output, ReLU is used where the number of outputs corresponds to the number of classes, and a softmax nonlinearity is used to provide probabilistic output. 

## 3. Results

### 3.1. Results for Sydney Urban Data Set

To test the accuracy of VoxNet CNN on the Sydney Urban Data Set, first, the parameters that would yield the best results were found. The best results occurred after training for 10 epochs with batch size 12. The accuracy, when tested against the test set (25% of the original data set), was 67.2%. Recall this data was very sparse, with the average density around 60 points/m^2^.

Then the data was down-sampled to determine the effect on the accuracy. Table 3 shows the accuracy was significantly affected.

### 3.2. Results for RedTail RTL-450 Data Set

Next, the VoxNet CNN was tested with the high-resolution data in the RedTail Data Set. The average data resolution for this set was over 700 points/m^2^, or more than 10 times that of the Sydney Urban Data Set. The data were down-sampled to 100%, 75%, 50%, 25%, 10%, and 5% of the original. These resolutions can be associated with data collection by the RedTail sensor at various altitudes and speeds of the UAV in Table 1. For example, the 75% data resolution (~562 points/m^2^) is comparable to an altitude of 120 m with a speed of 4 m/s, an altitude of 80 m with a speed of 6 m/s, an altitude of 60 m with a speed of 8 m/s, or an altitude of 40 m with a speed 12 m/s. The 50% density data would have about 375 points/m^2^, 25% about 187 points/m^2^, 10% about 75 points/m^2^, and the 5% about 37 points/m^2^. There are comparable numbers for each of these in Table 1, so you can determine the altitude and speed associated with these densities.

We trained VoxNet with each data density and then tested on each. Each model was trained for 8 epochs using a batch size of 32. This was to determine how accurate the classification could be if the data collected were sparse. Figure 6 shows the accuracy of classification for the down-sampled point clouds. The lines in the figure are each of the testing set densities shown by the series. The classification accuracy of each level of the down sampled training set is evaluated against each level of the testing set.

It can be seen from the graph that when the training set has a resolution of 100% of the original data collected by the sensor, we have very high accuracy (between 0.9957 and 0.9701). This indicates that when we have higher resolution data to train, testing data or even data collected in the field that is low resolution can still produce accurate classifications. As the resolution of the training data goes down, the classification accuracy also goes down. A resolution of 1% of the original data set would be equivalent to having about 7–8 points/m^2^ on the object.

Since the accuracy for 5% testing data was still relatively high, further testing was done to see how low the resolution of the testing data could go before the accuracy was affected. In Figure 7, it can be seen that as the training set resolution is reduced to 1% (still leaving the training set at 100% resolution), the accuracy is still around 0.9 (or 90%). However, the Model Loss (the error accumulation between the training data and the testing data) increases. Thus, the classification is not seriously affected, but the difference between the models is greater as we get closer to the 1%.

Additionally, note that the resolution of 10% of the original RedTail dataset is around 70 points/m^2^, approximately equal to the average resolution of the Sydney Urban Objects Dataset (61.2 points/m^2^), and the two separately trained models achieve nearly identical performance.

## 4. Discussion

When starting with the sparse Sydney Urban Data Set, the classification results of the VoxNet CNN were not good even on the full data set and only got worse when the data was reduced. For the denser data in the RedTail Data Set, the results of the data reduction study showed that if the training set is high resolution (100% of original), excellent classification results can be achieved even with sparse data as input. Lui et al. [11] showed in their data reduction study that there is no significant difference in DEM accuracy if data points were reduced to 50% of the original point density, and this reduction cut processing time in half. Similarly, our study shows that when the model is built on high-resolution data (100%), high accuracy can be achieved (around 90%) for testing data as low as 1% of the original data set’s point density. In our case, this does not affect processing time since training can be done a priori, and the classification results can be obtained quickly in all cases. The original VoxNet [9] paper reports classification takes around 6ms on a Tesla K40 GPU, and this time is constant since point cloud objects are converted to voxel format before classification. On an Intel Core i7 CPU, converting a point cloud of 30,000 points to voxel formats took 60ms using Python. Converting point clouds to voxel format is O(n) complexity.

The results also show that with sparse data to train on, if the input data is of about the same density, good classification accuracy can still be achieved. Peng et al. [12] concluded that the lowest LiDAR point cloud density required for accuracy for tree height extraction was 17 points/m^2^. However, for more complicated object classification, our research found that when training and testing with sets at 25% (187 points/m^2^) of the original data density, 89% accuracy could be achieved, although below that, the accuracy dropped below acceptable levels.

## 5. Conclusions

It should be noted that the RedTail LiDAR system can collect high-density data to be used as training data. The research also indicates that simulated data can be constructed to supplement training sets. The processing time of the high-density training data does not affect classification since the training is done a priori. The trained model can then be used during implementation for field classification. 

These findings have implications for classification during flights required to fly high and fast. These results can assist in making important operational decisions and be used for planning purposes. If flying high and fast is not important to the mission, then classification can reach near 100% for field data with high density.

## Figures and Tables

**Figure 1 sensors-22-05152-f001:**
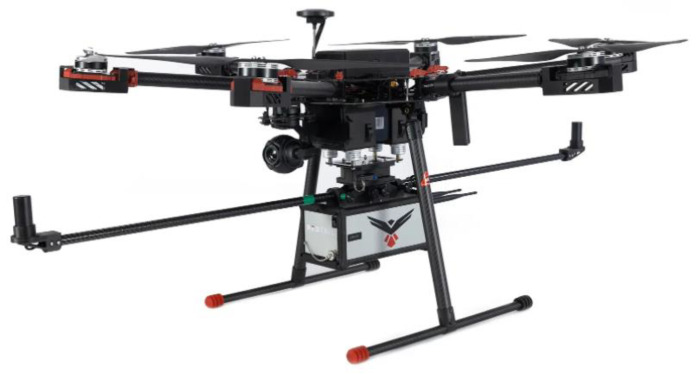
RedTail RTL-450 LiDAR on a Quadrotor UAV.

**Figure 2 sensors-22-05152-f002:**
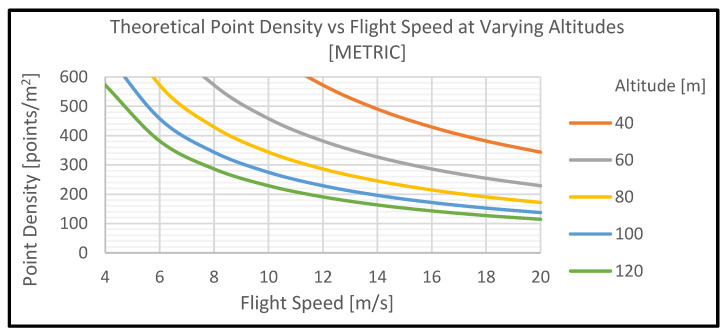
Theoretical point density, based on altitude and flight speed.

**Figure 3 sensors-22-05152-f003:**
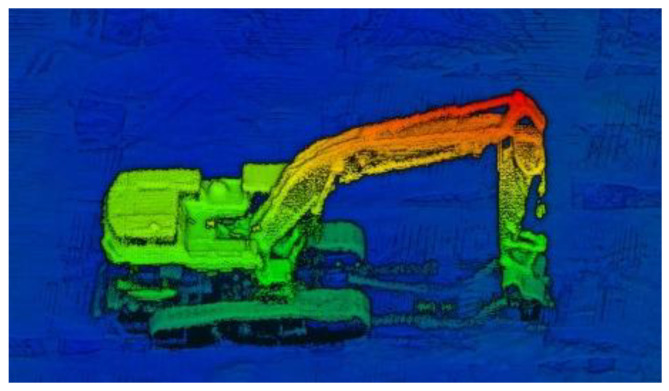
The image shows an example of a high-density point cloud collected by the RedTail LiDAR Systems RTL-450 sensor. Flight Specifications: altitude 30 m, speed 5.8 m/s. Coloring of the point clouds is by altitude.

**Figure 4 sensors-22-05152-f004:**
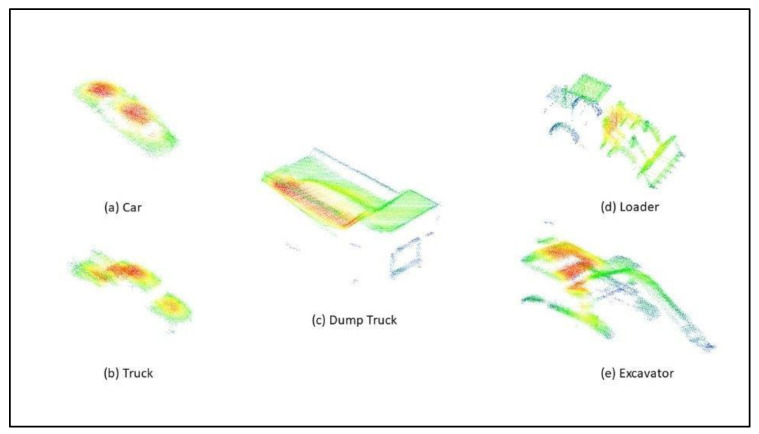
The images show an example of the point clouds for the five objects: (**a**) car, (**b**) truck, (**c**) dump truck, (**d**) loader, and (**e**) excavator. Coloring of the point clouds is by surface density.

**Figure 5 sensors-22-05152-f005:**
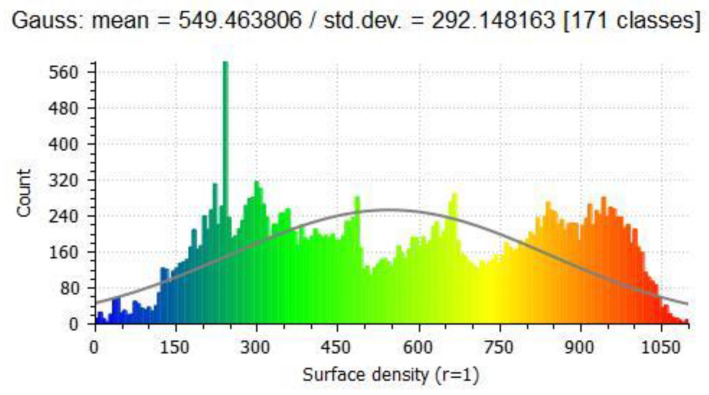
Histogram of surface densities at each point contained within the excavator point cloud is located in Figure 4. The *x*-axis is surface density, and the *y*-axis is the number of points on the excavator with that surface density. The colors depict the different densities with blue being least dense to red most dense.

**Figure 6 sensors-22-05152-f006:**
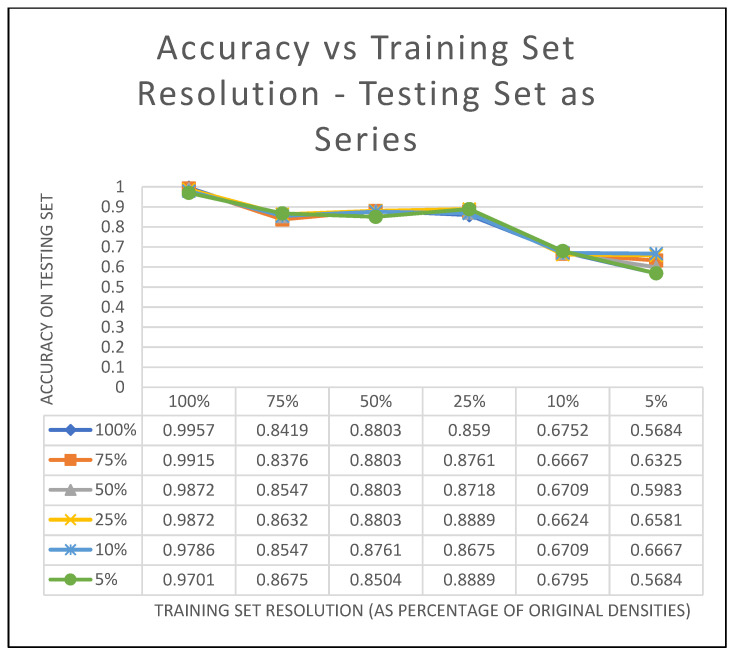
Each series in the graph is a different testing set resolution. The training set resolution is on the horizontal with the percentages shown as categories, and the accuracy of the testing set is on the *y*-axis.

**Figure 7 sensors-22-05152-f007:**
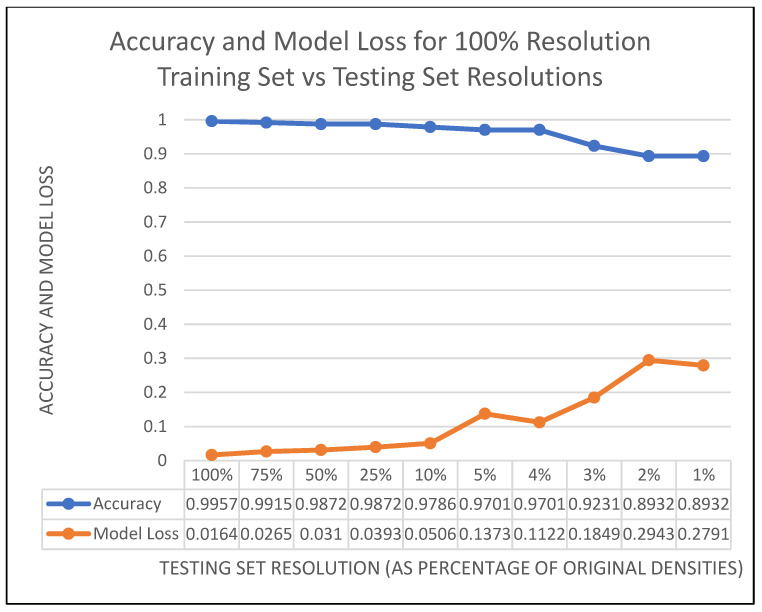
Accuracy and model loss on a model trained on 100% resolution data. The horizontal shows the testing set resolution as categories, and the *y*-axis is both accuracy and model loss on the testing set.

**Table 1 sensors-22-05152-t001:** Theoretical point density in points per square meter (points/m^2^).

Altitude (m)	Flight Speed (m/s)								
	**4**	**6**	**8**	**10**	**12**	**14**	**16**	**18**	**20**
**40**	1717	1145	859	687	572	491	429	382	343
**60**	1145	763	572	458	382	327	286	254	229
**80**	859	572	429	343	286	245	215	191	172
**100**	687	458	343	275	229	196	172	153	137
**120**	572	382	286	229	191	164	143	127	114

**Table 2 sensors-22-05152-t002:** Average point density in points per square meter (points/m^2^) for the training set.

	Car	Trucks	Dump Trucks	Loaders	Excavators
Representative Surface Density (points/m^2^)	762.5	787.2	705.0	761.8	722.8

**Table 3 sensors-22-05152-t003:** Accuracy of classification on down sampled Sydney Urban Data Set.

Data Resolution	100%	75%	50%	25%
**Accuracy**	0.6718	0.3029	0.2413	0.1753

## Data Availability

Not applicable.

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
