# Peer review of "Optimal LiDAR Data Resolution Analysis for Object Classification"

_sensors, 2022, doi:10.3390/s22145152_

Round 1
Reviewer 1 Report
This manuscript studied the effect of the LiDAR data resolution on the accuracy of object classification, aiming to obtain the optimal resolution. The topic is interesting, however, the method is not described clearly, so I can’t see its innovation. I addition, the third-party software CloudCompare was heavily relied on, and the specific operation steps of CloudCompare were introduced many times, making the manuscript look like a technical report. So, I have to reject it. The following are some specific comments.
(1) The standardization of writing needs to be improved.
-a) CNN was mentioned many times but in different forms, including “convolutional neural networks (CNNs)”, “Convolutional Neural Network”, “Convolutional Neural Networks”.
-b) There are also many expressions of point cloud density, such as “points per square meter”, “ppsm”, and “pt/m2”, which need to be unified.
(2) Experimental data are insufficient, more experiments need to be added.
Author Response
Added a paragraph at the end of the Introduction and added a paragraph at the beginning of the METHODS section to explain the methods employed and discuss the innovation compared to other papers referenced.
Moved section about data set resolution study from results to methods and added section to methods about comparison data (Sydney Urban Data Set).
Removed all but initial reference to CloudCompare, just to state what we used to perform the calculations.
Corrected all point density units to points/m2
Corrected later instances of “Convolutional Network Network” to “CNN”.
We added experimental results from another training/testing dataset, the Sydney Urban Objects Dataset, which is significantly sparser than the RedTail dataset we used.
Reviewer 2 Report
The manuscript discusses the resolution needed to classify 3D objects accurately and outlines how this resolution is accomplished for the RedTail RTL-450 LiDAR 12 System.
In practical cases, the change in resolution occurs due to a change in the environmental condition or the speed and distance. Is there any way to correlate resolution with the environmental conditions?
# The computation complexity of the proposed method should be clearly described.
# Classification testing should be done for real-time experiments. Moreover, a comparison between the proposed work and other works is required.
# If you are citing the other's work, their contribution(s) has to be written in your own words.
# Many important research demonstrating similar research have not been considered. For example,
- Gao, H., Cheng, B., Wang, J., Li, K., Zhao, J., & Li, D. (2018). Object classification using CNN-based fusion of vision and LIDAR in autonomous vehicle environment. IEEE Transactions on Industrial Informatics, 14(9), 4224-4231.
-He, X., Wang, A., Ghamisi, P., Li, G., & Chen, Y. (2018). LiDAR data classification using spatial transformation and CNN. IEEE Geoscience and Remote Sensing Letters, 16(1), 125-129.
Author Response
Added the following statement to the introduction to clarify: “It should be noted that we do not consider environmental factors in this paper, since when using a small UAV as pictured in Figure 1 for data collection, conditions must be favorable (low wind and no rain) before flights are attempted.”
Added computational complexity of data processing to Section 2.2, but not classification as the input size is fixed, which means that classification time will be hardware dependent.
Added the following statement to the introduction to clarify: “It should be noted that we do not consider environmental factors in this paper, since when using a small UAV as pictured in Figure 1 for data collection, conditions must be favorable (low wind and no rain) before flights are attempted.”
Added computational complexity of data processing to Section 2.2, but not classification as the input size is fixed, which means that classification time will be hardware dependent.
We currently do not have a real-time system in place for testing. All instances of real-time have been reworded to be more reflective of the state of the current research.
Reworded each of the sections mentioned.
Added sentences to methods referring back to reduction studies.
Although the Gao et al. paper has impressive results, it relates to autonomous ground vehicles and data fusion between visual data and LiDAR. They do utilize a CNN, but I do not think the work is relevant to our paper.
We included the He et al. paper in the introduction and also included another paper that was also cited by He et al.
- Jaderberg, K. Simonyan, A. Zisserman, and K. Kavukcuoglu, “Spatial transformer networks,” in Proc. Adv. Neural Inf. Process. Syst., vol. 25, 2015, pp. 2017–2025.
Reviewer 3 Report
This is an interesting paper considering very relevant field of LIDAR for 3D positioning. Authors consider using convolutional neural networks to process lidar data with decreasing amount of detail.
It would be very interesting if authors would discuss possibility of applying sparse methods (like sparse tensor structures etc) for CNNs in order to exploit natural structure of pointcloud coming from lidar. Its a great potential for memory use reduction and acceleration of computation. While there are no mainstream implementations it is a rapidly developing field.
Author Response
Although applying sparse methods is an interesting idea, we felt that would take our research in a different direction and we do not feel that this is the direction we want to go with our study at this time, but will keep this in mind for future research.
Round 2
Reviewer 1 Report
All my concerns are sloved.
Author Response
Thank you